# Integrating Non-Destructive Surveys into a Preliminary BIM-Oriented Digital Model for Possible Future Application in Road Pavements Management

**Fabrizio D'Amico** [1,*], **Luca Bianchini Ciampoli** [1], **Alessandro Di Benedetto** [2], **Luca Bertolini** [1] **and Antonio Napolitano** [1]

1 Department of Engineering, Roma Tre University, 00146 Rome, Italy; luca.bianchiniciampoli@uniroma3.it (L.B.C.); luca.bertolini@uniroma3.it (L.B.); ant.napolitano@stud.uniroma3.it (A.N.)
2 Department of Civil Engineering, University of Salerno, 84084 Fisciano, Italy; adibenedetto@unisa.it
* Correspondence: fabrizio.damico@uniroma3.it

**Abstract:** The implementation of the digitalization of the linear infrastructure is growing rapidly and new methods for developing BIM-oriented digital models are increasing. The integration of the results obtained from non-destructive surveys carried out along a road infrastructure in a pavement digital model can be a useful method for developing an efficient process from a pavement management systems (PMS) point of view. In fact, several applications to optimize PMS have been thoroughly investigated over the years and several researchers and scientists have investigated significant elements for improving the PMS applied to a transport network, including road infrastructures. This study presents a new, tentative process for implementing into a BIM environment the dataset processed from two surveys carried out in a case study. Moreover, the main reason for this investigation is related to the need for an effective system able to evaluate continuously the pavement conditions and programming maintenance interventions. To date, both the instruments and the methods to detect the pavement configuration have evolved, along with the development of non-destructive technology (NDT) tools such as laser-scanners and ground-penetrating radar. Finally, the main results of the research demonstrate the possibility to provide a digital twin model from the synergistic use of geometric and design information with the results from monitoring conducted on a road infrastructure. The model can be potentially used in future BIM-based PMS applications.

**Keywords:** pavement management systems (PMS); non-destructive technology (NDT); horizontal building information modeling (BIM); laser-scanner (LS); ground penetrating radar (GPR); road pavement modeling; transport infrastructures monitoring and management; non-destructive analysis

## 1. Introduction, Main Objectives and Literature Review

As vehicular traffic over linear transport infrastructures is increasingly growing and is nowadays the structural integrity of their components is more demanding than ever, it seems imperative that road pavements receive constant and efficient management in order to maintain the required functional and safety standards for road users. Knowledge of the condition of the pavements is necessary for effectively planning the eventual maintenance activities, thereby allowing for the optimization of the available resources. In this regard, the spread of non-destructive technology (NDT) tools such as the mobile laser scanner (MLS) and ground-penetrating radar (GPR) nowadays permits more thorough and efficient surveys, without the need for compromising the integrity of the pavement structure.

In fact, several studies have demonstrated the effectiveness of both GPR and MLS as applied to the monitoring of bridges and roadways. Indeed, various successful applications were reported in the evaluation of the quality of the pavement of roadway infrastructure [1–3]

and in the effective representation of the geometry of even a single structural element with millimeter precision [4,5], respectively.

Once the needed data are acquired through NDT surveys, it is then possible to define and evaluate an optimized management program. Indeed, the information on the pavement that are extracted from the surveys represent a key point for planning and executing future maintenance and rehabilitation interventions.

Within this context, building information modeling (BIM) stands as a very useful methodology for increasing the efficiency of the management of the dataset provided by the NDT surveys on the pavement asset. As confirmation, the use of BIM in civil engineering is strongly supported by national and international experiences that aim at improving and optimizing the processes defining a civil work life cycle, which span from its design to the management phase [6–11].

In Italy, following the European Directive [12] that encourages and specifies the use of BIM for publicly funded construction and building projects in the European Union by 2016, BIM-based procedures have been progressively integrated by decrees of the Ministry of Infrastructure and Transport. The first one regarding these topics was published in 2017 [13] and, with its updates in 2021 [14], aims at a more efficient design of new civil works. Moreover, in response to the tragic events concerning the collapse of the "Morandi bridge" in the city of Genoa, new measures for both the safety of the national transport infrastructure network and the risk classification and management of bridges and viaducts were implemented [15,16]. These regulations state the BIM-based procedures to be considered for the management of pre-existing civil works, while still even though at the experimental level. Within this framework, the use of NDT techniques for assessing the current conditions of the inspected asset plays a crucial role.

In the literature context, only few studies have tackled the topic of the actual possibility to use and integrate data from NDT surveys for the BIM-based management of maintenance activities on pavements in road networks [6,17–21].

At the same time, over the years, pavement management systems (PMS) have been thoroughly updated and improved to ensure the endurance of the correct state of road pavements. This becomes a crucial topic for transport infrastructure, in airports for example, where pavements play a strategic role not only for maintenance purposes, but for the overall management of the asset. A review of the existing literature regarding the assessment systems and models for pavement conditions was sought by de Moura et al. in 2020 [22].

As a PMS generally consists of a set of procedures and tools designed to assist decision-makers in identifying the most appropriate strategies for monitoring and maintaining pavements in a proper condition, this study aims to define a possible procedure to implement BIM methodology for the possible future definition of a BIM-based PMS, in order to ensure that the data collected during the inspections can be stored in dedicated designed digital models of the pavement. In fact, these models may represent a starting point for analyzing the condition of the road in a more thorough and efficient way, as they can then be progressively updated at each new survey, thereby permitting pavement distress to be detected and to control their evolution over time, while being aware of the effect of any delay in maintenance activities. Moreover, the study hereby presented describes the process to build a digital model of a road pavement, that can store data provided by different surveys carried out on site. This is only a possible starting point for a future BIM-based pavement management system, as a PMS not only provides information regarding the road, but also includes pavement condition assessment, condition prediction, and best maintenance determination. Therefore, there is the will to analyze in the continuation of the study the different procedures in which the digital model hereby described could be used to face these other aspects of a PMS.

## 2. Methodology and Case Study

*2.1. Methodological Framework*

The possibility of integrating information into digital models is the fundamental assumption of the BIM methodology. These models need to store data and information in order to be subsequently analyzed, so that the current state of the infrastructure can be determined at each stage of analysis. Therefore, the components of the model must be able to receive input data from the operator who is carrying out the digitalization and to give back output information once the analysis of the model is performed. Figure 1 shows the methodological framework that was defined during the course of this study.

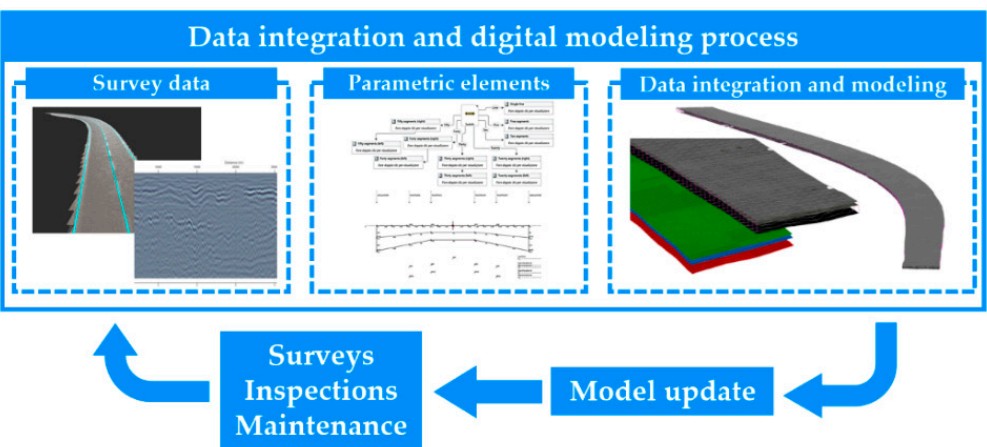

**Figure 1.** The methodological framework.

As a first activity, the management of survey data has to be carried out, in order to provide the necessary information to generate the digital model. The integration of the data regarding the infrastructure is made possible by the use of parametric elements that can adapt their features to the information provided by the previous activity. At last, the digital model is created, making use of these parametric components and the edited data extracted from the surveys.

The contents of the study that is presented in this article are part of a wider road pavement management process. This can be carried out by continuously updating the model relying on the same methodology hereby proposed, after each round of surveys or after maintenance operations are executed on the infrastructure.

*2.2. Case Study*

2.2.1. Site of Investigation

The experimental database used in this work derives from a wider survey campaign conducted on a transport network located in proximity to the city of Salerno, Campania Region, southern Italy, to evaluate its overall resilience towards natural hazards and, particularly, landslides.

Specifically, the experimental campaign included some NDT inspections, including GPR, MLS, synthetic aperture radar interferometry (InSAR) and laser profilometer. The transport network that was selected due to the known geomorphological issues affecting the whole area of Salerno included a motorway (A3 Salerno–Reggio Calabria), a two-lane rural road (SS18) and a double-track ballasted railway (Cava de' Tirreni-Salerno stretch). According to the main objective of the project, the comprehensive experimental campaign, geographically shown in Figure 2, involved a total length of surveys of about 20 km.

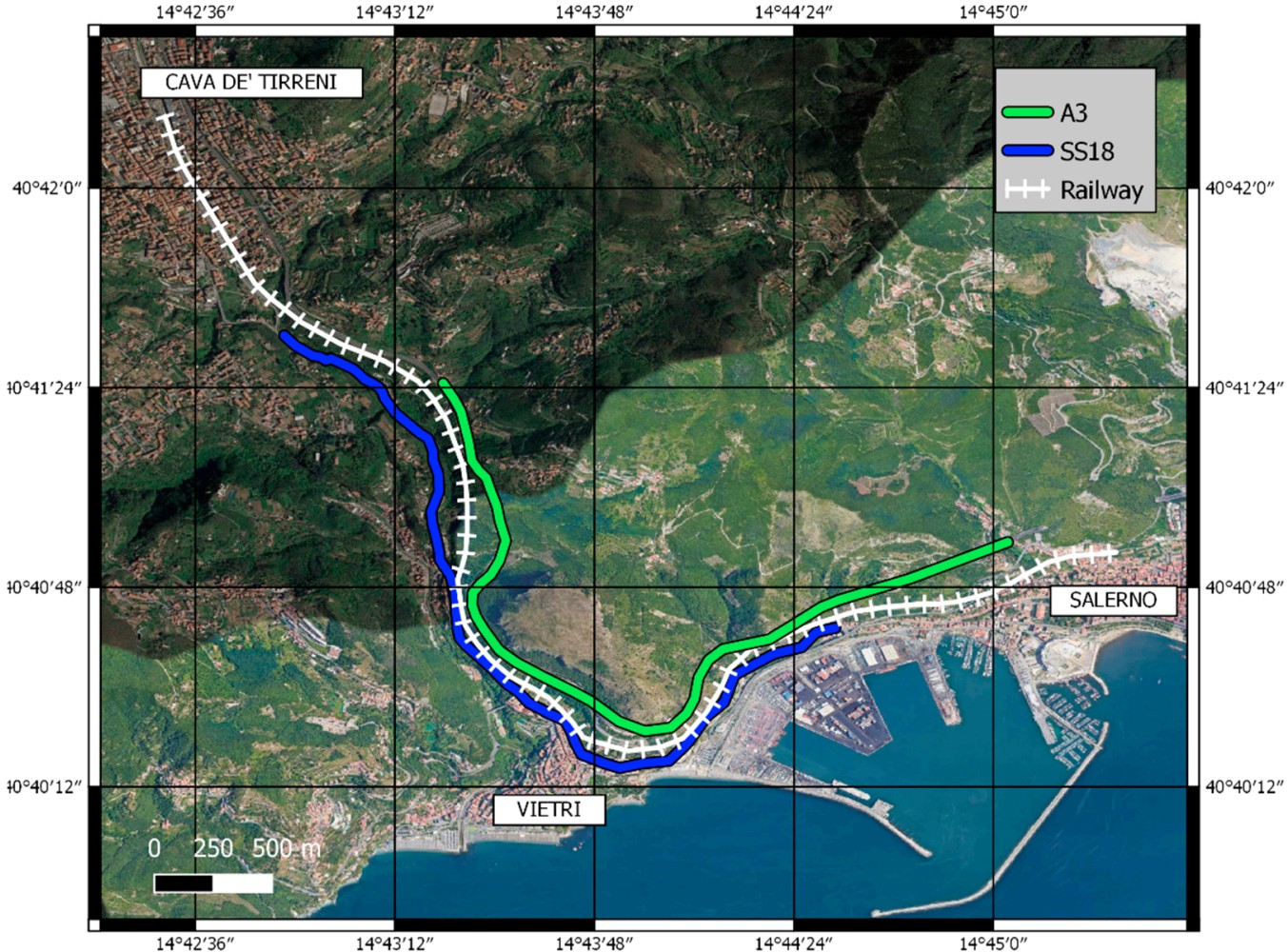

**Figure 2.** Geographical framework of the overall experimental campaign (Reference System: WGS84 UTM33N—Basemap: Google Satellite).

Concerning the present study, out of the total experimental database collected, only the GPR and the MLS datasets acquired on the 4.5 km-long A3 motorway stretch are considered (green line in Figure 2). Specifically, the motorway is composed of two carriageways composed of two lanes each. On average, the carriageways are composed of 3 m-wide lanes, with 0.50 m-wide shoulders on each side.

As in the example reported in Figure 3, the two carriageways follow a different alignment due to the complex orography of the area. Accordingly, a concrete retaining with a variable height that reaches up to 7 m works as a divider between the carriageways.

Parallel to this, the challenging morphology of the terrain have caused the presence of several engineering structures along the inspected stretch. In particular, along the surveyed length of the motorway, a total amount of seven Maillart-like bridges, one natural tunnel and two artificial tunnels are located.

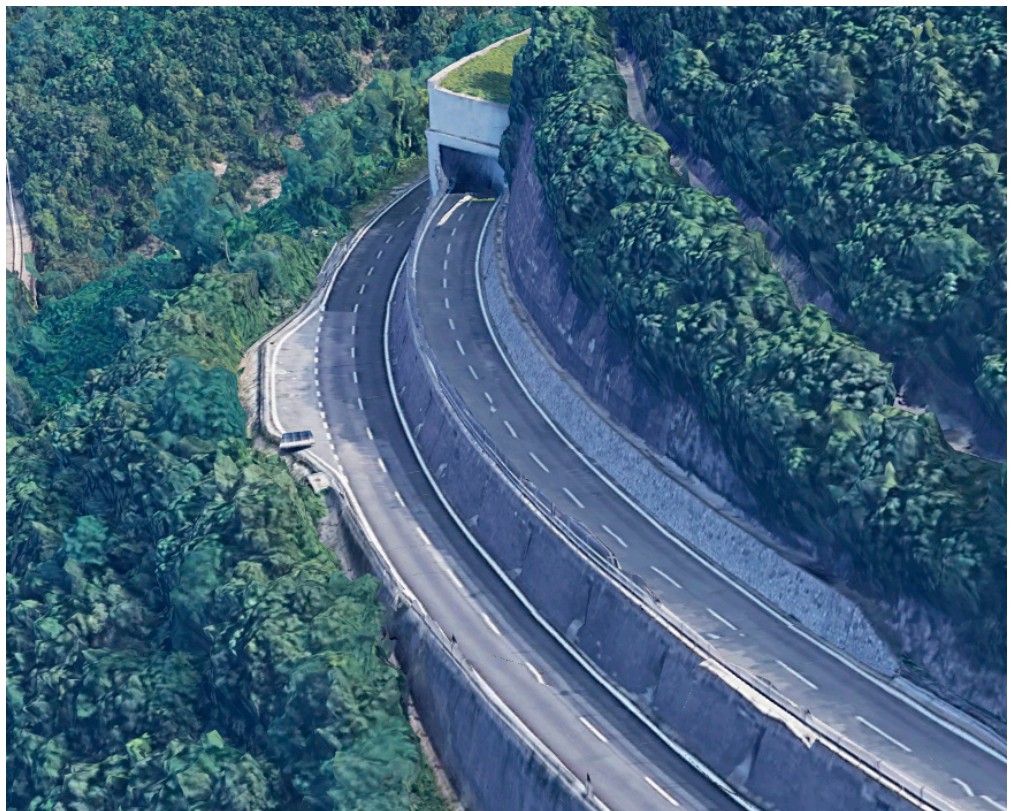

**Figure 3.** Example of the A3 motorway configuration along the inspected stretch (Source: Google).

2.2.2. Laser Scanner Survey

The mobile laser scanner (MLS) data were acquired by a mobile laser scanner (Road-SIT Survey, developed by Siteco Informatica s.r.l.), mounted on the roof of a vehicle driving on the A3 motorway. The measuring head is equipped with two laser scanners Faro CAM2 Focus, symmetrically configured on the left and right sides, pointing to the rear of the vehicle with an inclination angle of approximately 45°. This configuration is called the 'X' pattern. The laser's acquisition frequency is 976 kHz, with a mirror speed of 96 Hz.

The vehicle travelled a number of times at different average velocities (20, 50, 70, 90 km/h), with a constant acquisition frequency (976 kHz). The output data are point clouds (Figure 4a) directly georeferenced in the Italian cartographic system (UTM33/RDN2008).

The resulting scan lines have an angular range of 320°, and a distance range of 80 m. The distance between the scanning lines is approximately 3.5 cm and 11 cm, respectively for driving speeds of 20 and 90 km/h.

Figure 4b shows for each velocity an excerpt of the density map represented through a chromatic scale that shows how the density longitudinally to the path is almost constant while it ranges greatly with the velocity (from 18,000 points per $m^2$ to 4000 when the velocity increases from 20 to 90 km/h).

Follow-up analyses were made on the less dense cloud (about 4000 points per $m^2$), the one acquired in only one travel direction, at a driving speed of 90 km/h. Anyway, the density of this point cloud allows for accurate modeling of the infrastructure.

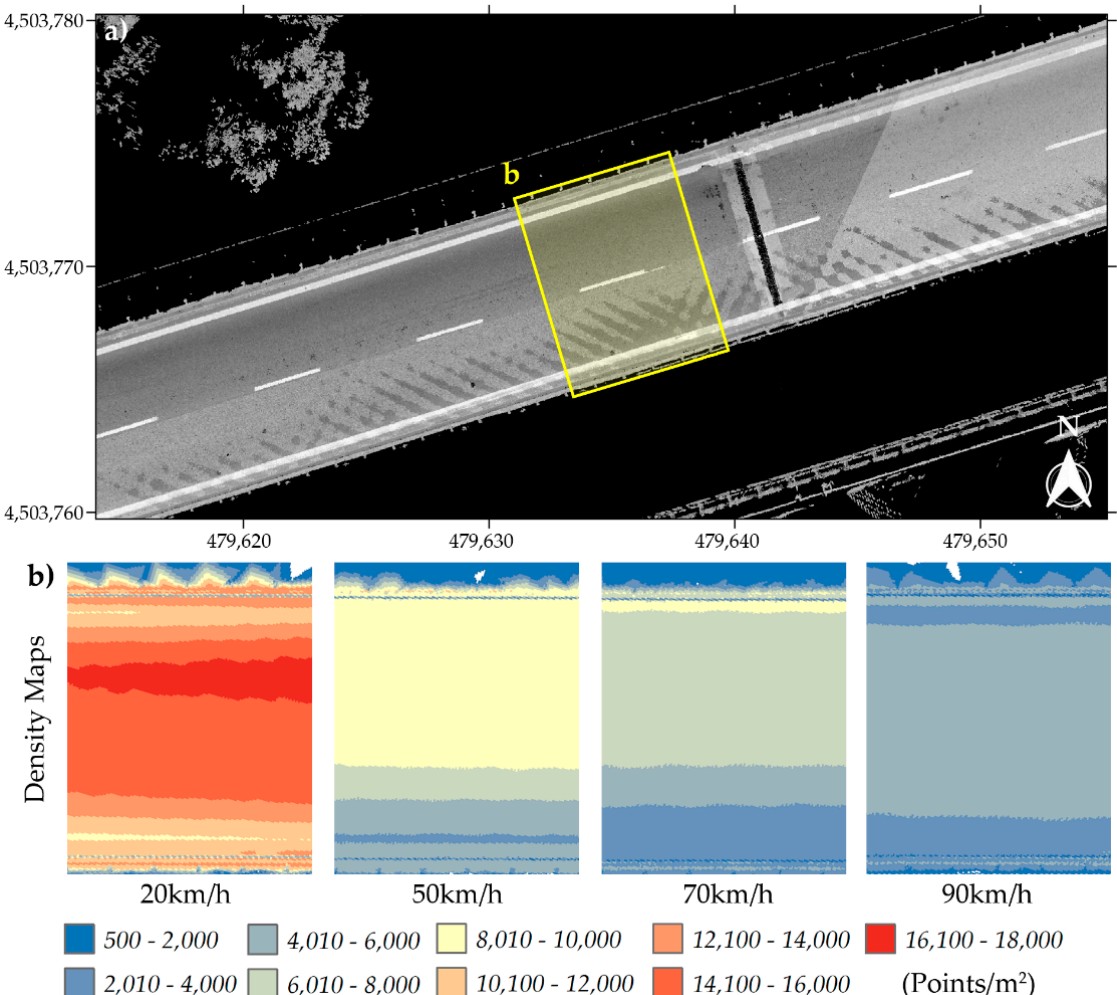

**Figure 4.** Density map. (**a**) Top view of the excerpt of the point cloud of the single carriageway, the color scale is a function of the intensity values; (**b**) density maps at different average velocities (20, 50, 70, 90 km/h).

### 2.2.3. Ground Penetrating Radar (GPR) Survey

The dataset was collected through a GPR system equipped with an air-launched horn antenna having a central frequency of 1000 MHz. The acquisition system was mounted onto a survey vehicle through a support including a suspended beam that allowed the antenna to work at the correct height from the pavement surface. The experimental set-up is shown in Figure 5, while Table 1 reports acquisition parameters.

**Table 1.** GPR acquisition parameters.

| Parameter | Values |
| :---: | :---: |
| Central frequency | 1000 MHz |
| Time Window | 25 ns |
| Horizontal resolution | 0.05 m |
| Samples | 512 |
| Acquisition speed | 50–70 km/h |

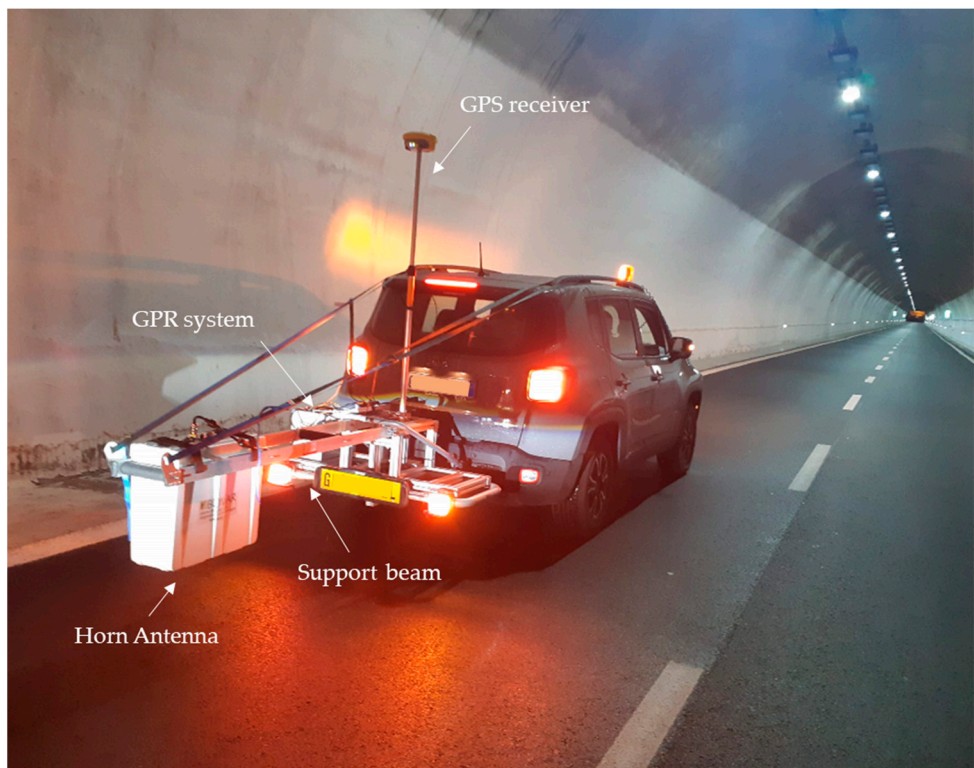

**Figure 5.** The acquisition setup used for ground penetrating radar (GPR) surveys.

The survey has been conducted by travelling along the motorway stretch at different lateral positions within the carriageway. Even though the motorway had been closed to the traffic to support the efficiency and safety of the survey operations, the acquisition speed adopted for GPR tests was reported to comply with the presence of traffic on the road (i.e., 50–70 km/h). The survey protocol, which included two scans for each lane composing the carriageway is described in Figure 6. Due to the active worksites located along its track, it was possible to perform only a single acquisition on the north-bound carriageway.

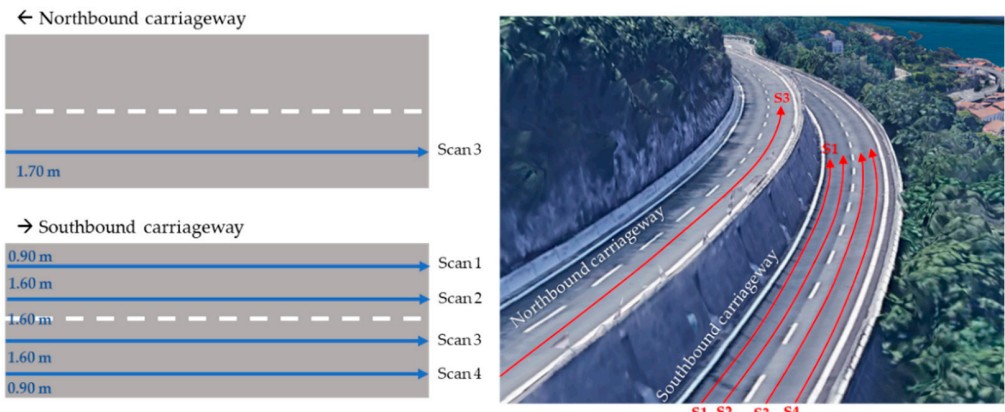

**Figure 6.** The scheme of acquisition adopted for GPR surveys.

Lastly, each GPR scan was run in parallel to a GPS record, which allowed georeferencing the GPR traces in the Italian cartographic system (UTM33/RDN2008).

## 3. Results on the Creation of the Digital Model

Using the non-destructive surveys previously described, a digital model of the road superstructure was created. In this first preliminary application, the modeling process was

conducted for the southbound carriageway as a denser survey database was available, especially regarding the inspections conducted with the GPR.

The workflow for creating a digital model is shown in Figure 7.

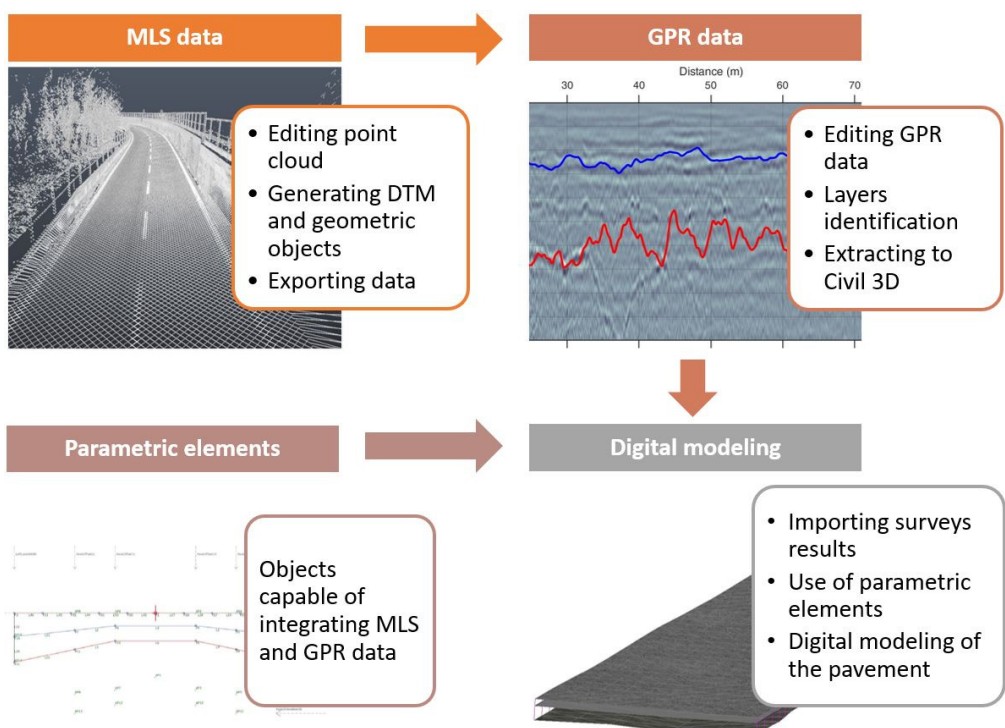

**Figure 7.** Proposed workflow for the integration of non-destructive survey data in a building information modeling (BIM) model.

### 3.1. Mobile Laser Scanner (MLS) Data Inputs

The first step in the process was to operate on the point cloud obtained from the mobile laser scanner so that it would be more manageable in the subsequent phases. From the same point cloud, through the use of the Autodesk Infraworks software [23] georeferenced polylines representing the road markings on the roadway were extracted. To achieve said result a specific feature of the software was used, as it can recognize different intensity areas within the point cloud, which differs depending on the types of surface that are detected during the laser scanner survey. Once these areas are identified, linear objects are automatically drawn along them, resulting in the previously mentioned three-dimensional polylines. At this point, the feature polylines of the carriageway's axis and the margins of the lanes have been extracted. Another important use of the point cloud is to generate a three-dimensional surface that represents the surface characteristics of the pavement. The need to integrate further data to define a three-dimensional modeling arises, to characterize the thicknesses and configuration of the deep layers of the pavement.

### 3.2. GPR Data Inputs

In order to reliably extract from the GPR dataset useful information concerning the pavement structure, the data have been processed according to the most advised procedures in the literature [24,25]. In particular, key information on the development of the presented approach is the layering configuration of the pavement. Repeated resurfacing interventions, low dielectric contrast between layers (e.g., base to subbase) and wrong survey configuration (e.g., central frequency of application) are factors that may inhibit a clear recognition of the layers. To avoid any potential misinterpretation of the data and, accordingly, a wrong reconstruction of the pavement configuration, more information

on both the pavement design and previous maintenance activities, and the extraction of corings along the surveyed stretch are always suggested.

In the present study, in addition to the design charts provided by the administrator of the road network, a total of three cores for each carriageway have been extracted from the pavement for both verifying the interpretation of the data and calibrating the speed of propagation of the EM waves used for calculating the depth of the layers.

In fact, starting from the receiving time of the reflections in the processed data, the depth of the layers' interface was obtained by comparing the GPR signal collected in correspondence to the position of pavement corings. As a result, for each GPR scan, it was possible to recognize the configuration of the pavement layers, as shown in the example in Figure 8a.

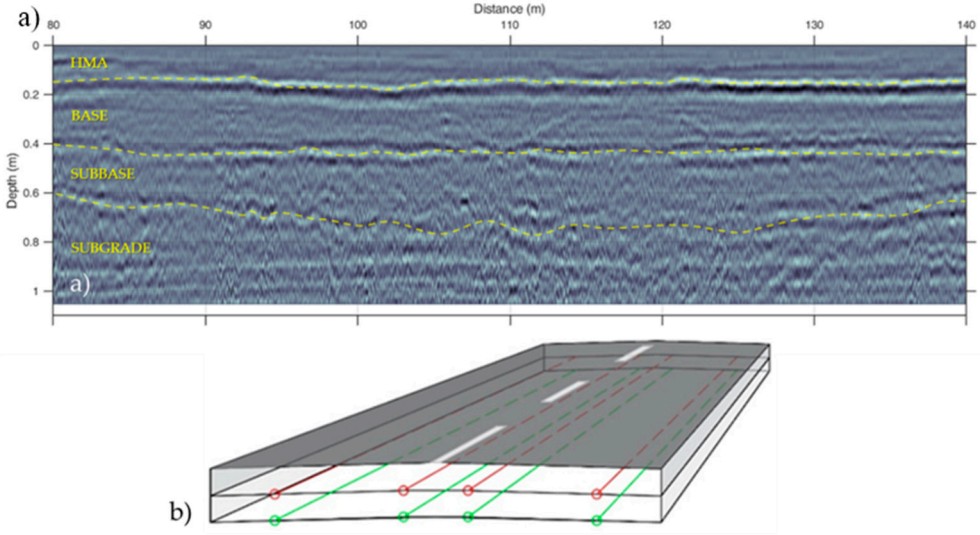

**Figure 8.** (**a**) example of GPR scan extracted by the survey dataset with evidenced the interfaces between the layers; (**b**) 3D polylines generated from the management of GPR data.

As a further step, by managing the GPR database that was georeferenced by parallel GPS acquisitions, a series of points characterized by two coordinates x and y corresponding to their latitude and longitude and a z referring to the depth of the layers interface have been obtained. By connecting those points, for each GPR trace, a pair of three-dimensional polylines were produced, referring to the two layers of the pavement currently modeled (see Figure 8b). In particular, it was always possible to analyze the interface between the hot-mixed asphalt (HMA) and the base course (red line in Figure 8b) and the one between the subbase and the subgrade (green line in Figure 8b). In fact, due to the low dielectric contrast between base and subbase, it was not possible to reconstruct the interface between these layers for the whole length of the inspected stretch.

### 3.3. Creation of Parametric Elements

The starting point for the digitization process is the creation of parametric elements able to integrate the data assigned to them, in order to generate a three-dimensional model. In the case study, a parametric road cross-section was created, able to integrate the information extracted from the laser scanner and GPR surveys.

The polylines extracted from the point cloud work as targets for the cross-section's axis and its edges in the direction of travel, so it can adapt its dimensions to the width of the analyzed highway, modeling the area of the pavement that is generally carriageable.

A further parameter required for the functioning of the parametric section is the three-dimensional surface generated by the point cloud obtained from the laser survey. Making use of a certain number of segments for each lane, this surface can be reproduced with an adequate level of approximation within the model (purple lines in Figure 9). The

said number can be chosen during the modeling phase. This can vary from a single line for each lane to 100 segments for the whole carriageway, depending on what the model is going to be used for, as its computational complexity varies along with the precision used to recreate the pavement. This allows a digital reproduction of the infrastructure to be obtained characterized by a certain surface irregularity that approximates the one present on-site. Another use of this surface is to provide the reference from which to determine the depth of the superstructure layers.

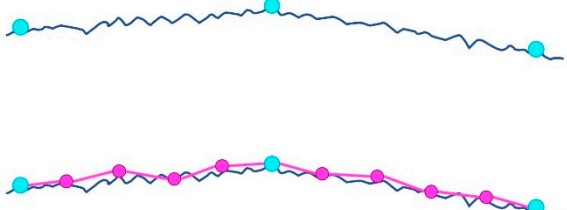

**Figure 9.** Example of the approximation method for the pavement surface.

The next step is to integrate the data generated by the GPR survey into the digital model. Following the proposed methodology, said data comes in the form of eight three-dimensional polylines that correspond to eight points in the parametric cross-section. By joining these points, the configuration of the deep layers of the superstructure can be determined. The links that unite these points are coded in relation to the level they represent in the package of the superstructure (see Figure 10).

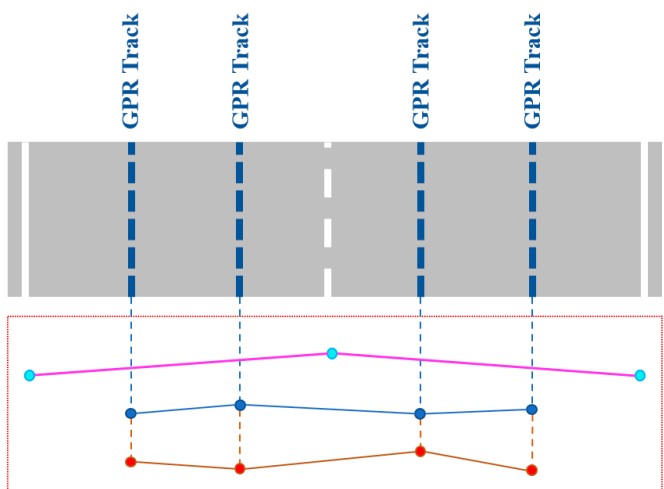

**Figure 10.** Implementation of GPR data in the parametric section.

Thanks to the available dataset it was then possible to model the surfaces representing the deep layers of the pavement, with some necessary approximations regarding its depth underneath the edges of the lanes, where GPR data have not been collected. A linear interpolation of the points determined from the GPR data is used to define the configuration of the pavement in the most external parts of the carriageway. By conducting a GPR survey using a hand-towed ground-coupled radar this kind of approximation could be avoided, since that instrument allows information to be collected in correspondence of the edges of the road, being not fixed to the survey vehicle.

The parametric cross-section hereby described is then able to adapt both its width and thickness to the information provided by the non-destructive surveys, so that once it is extruded along a specified alignment it generates a model of the analyzed road.

### 3.4. Digital Modeling

To create the necessary model of the analyzed road, Autodesk Civil 3D was used [26]. An alignment was created using the central three-dimensional polyline extracted from the point cloud, so that the road model would follow both the planimetric and altimetric features of the existing road. Once the parametric section was assigned to it, it was then extruded along its axis line, thereby generating a three-dimensional model of the pavement whose dimensions vary in relation to the 3D polylines extracted from the surveys. The model is configured as three superimposed planes, which together form the package of the superstructure, as reported in Figure 11 below. The proposed methodology then allows for the creation of a new road model by relying only on data generated from surveys carried out on site. Moreover, specific information regarding the real infrastructure present on site can be assigned to the model, such as the materials that constitute the road superstructure or the direction of travel of the modeled carriageway.

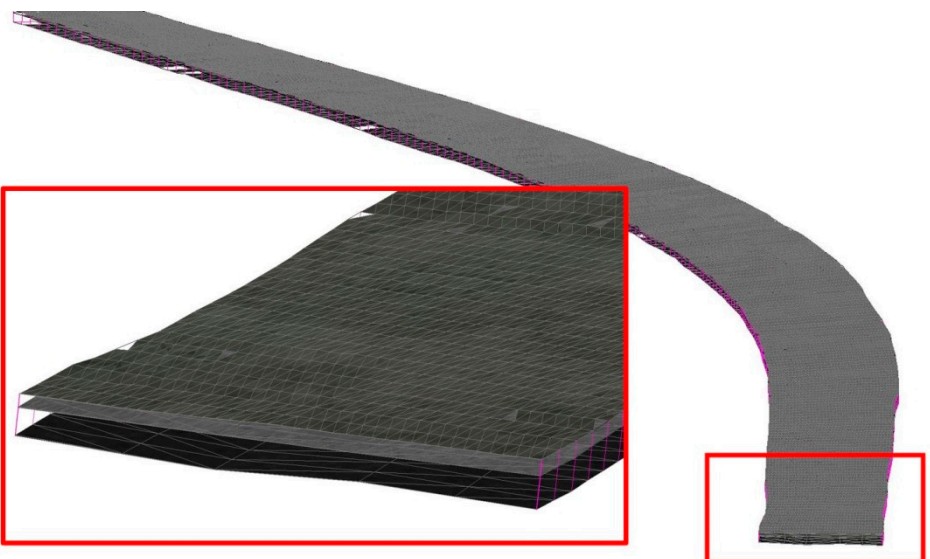

**Figure 11.** The pavement digital model.

The application used allows three-dimensional surfaces to be created starting from the coded links present inside the model. As has been previously illustrated, each layer was assigned a specific code so that a different surface for every layer of the pavement could be generated. A first analysis was carried out to evaluate the trend of the slopes of these surfaces, so that the possible outflow of the platform waters and those that reach the depth of the pavement could be established (Figure 12a). Using the same information, drainage basins for each layer of the pavement could be determined. This allowed the areas of the infrastructure to be defined where the water was not correctly drained to the side of the carriageway as supposed (red areas in Figure 12b), which may represent a risk for the structural integrity of the road as these problems occur in the deep layers and for the security of the road users, especially motorists when it happens on the surface level. These evaluations are currently limited by the approximations made when coding the parametric section. In fact, the central area of the roadway, which was analyzed by the survey instruments, is more faithful to the real configuration present on site.

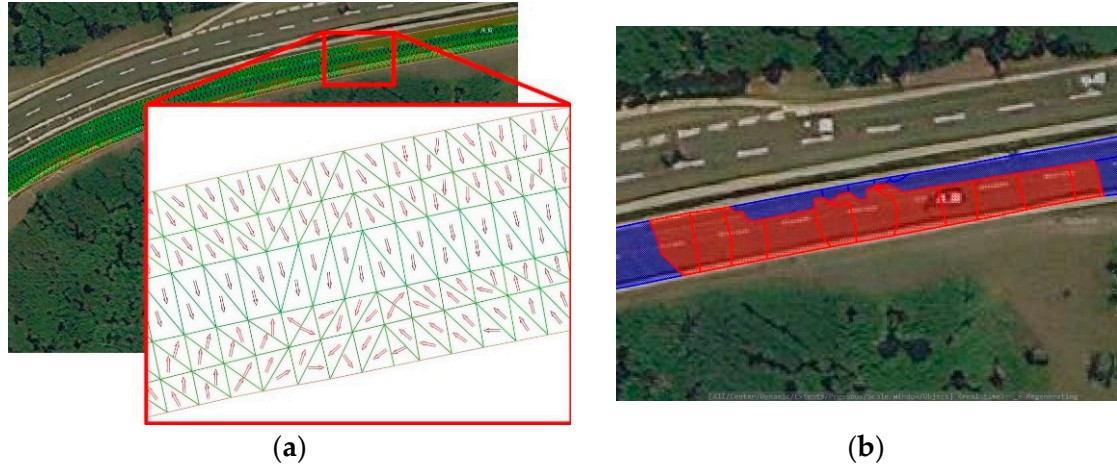

(**a**)        (**b**)

**Figure 12.** Analysis results on the digital surfaces generated by the model: (**a**) slope patterns in a certain area of a layer of the pavement; (**b**) drainage basins on a layer of the pavement with an indication of those that form a depression on the surface.

### 4. Conclusions, Challenges and Future Perspectives

The main objective of this study was to define a first potential methodology to integrate different datasets generated from non-destructive surveys carried out on a linear transport infrastructure within a BIM model. Making use of different software programs the integration between MLS and GPR data was successfully carried out, generating a digital model of the pavement of a road. The point cloud obtained from the laser scanner allowed the geometric definition of the model and its position in a georeferenced system. The data generated by the GPR was used to determine the thickness of the pavement layers, allowing the model to be defined in three dimensions. The resulting digital model was used to carry out useful analysis regarding the state of the superstructure, even in areas that usually cannot be reached by operators.

The integration of the results obtained from non-destructive surveys carried out along a road infrastructure in a pavement digital model can be a useful method for developing an efficient process from a PMS point of view. Moreover, the model can be potentially used in future BIM-based PMS applications.

At the current stage, the study represents a promising starting point that is, however, effective in stressing the potential of the approach. As the study hereby presented represents a preliminary approach to the implementation of BIM processes in the management phase of a road pavement, it has some limitations and challenges. In detail, more information regarding the condition of the pavement's surface should be implemented in the model, as it only manages to describe its irregularity by approximating the point cloud. As this approximation is executed by dividing each section of the road in a maximum of 100 segments, a more thorough analysis of this characteristic of the pavement could be carried out. Another current limitation is the absence of data regarding the deep layer of the superstructure on the most external parts of the carriageway, as the GPR could not reach those areas. Another approximation had to be made to determine the configuration of the layers underneath the edges of the lanes. These limitations could affect the analysis carried out on the model, therefore more studies and applications are currently being examined to optimize the proposed process in the next stages of the research.

Possible implementations that would improve the model accuracy in representing on-site conditions are subsequently described. First of all, the model should be able to represent the entire road body, thus extending beyond the limits dictated by the horizontal signage. To do this it is necessary to determine new targets to be assigned to the parametric cross-section, which can be automatically extracted from the surveys carried out on site.

An improvement that can be made to the model as currently composed is the inclusion of data from a hand-towed ground-coupled radar that, as specified above, would allow

the depth of the road pavement to be determined faithfully along the entire roadway. This would require the creation of a new parametric section, able to describe the superstructure more accurately.

Other integrations could concern the modeling of the elements of road furniture located around the roadway and already present within the mobile laser survey. Digitizing these elements would increase the level of detail of the BIM model, allowing an integrated management of the various features of the infrastructure, employing the concept of digital twins, that could be used for the maintenance workflow.

**Author Contributions:** Conceptualization, F.D. and L.B.C.; methodology, F.D., L.B.C., L.B. and A.N.; investigation, F.D., L.B.C. and A.D.B.; data curation, L.B.C., A.D.B., L.B. and A.N.; writing—original draft preparation, F.D., L.B.C., A.D.B., L.B. and A.N.; writing—review and editing, F.D., L.B. and L.B.C.; project administration, F.D.; funding acquisition, F.D. and L.B.C. All authors have read and agreed to the published version of the manuscript.

**Funding:** The research is supported by the Italian Ministry of Education, University and Research under the National Project "Extended resilience analysis of transport networks (EXTRA TN): Towards a simultaneously space, aerial and ground sensed infrastructure for risks prevention", PRIN 2017, Prot. 20179BP4SM. In addition, the authors acknowledge funding from the MIUR, in the frame of the "Departments of Excellence Initiative 2018–2022", attributed to the Department of Engineering of Roma Tre University.

**Conflicts of Interest:** The authors declare no conflict of interest.

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
