# Peer review of "Integrating Non-Destructive Surveys into a Preliminary BIM-Oriented Digital Model for Possible Future Application in Road Pavements Management"

_infrastructures, doi:10.3390/infrastructures7010010_

Round 1
Reviewer 1 Report
- Please suggest the information input to the current pave management system and the requirements for improvement to the BIM-oriented digital model.
- It is known that it is very difficult to distinguish soil profiles with ground-penetrating radar. Please suggest whether the base layer, sub-base layer, subgrade, and ground under the pavement can be distinguished by GPR.
- I know that road friction coefficient and soil support value are essential information in the pave management system, but it is necessary to obtain this information with a laser scanner and GRP and explain whether it is possible to input it into the BIM model.
Author Response
Dear Reviewer,
we are pleased to submit the new revised version of the paper “Integrating non-destructive surveys into a BIM-oriented digital model for road pavements management” that addressed your useful remarks, and the point-by-point response to your comments.
In fact, based on your comments, we have had the opportunity to examine more in-depth our paper and our work in general. We have considered in depth all the useful comments made by you and by the other reviewers. Below you can find your point-by-point response and, in the new revised paper we have included all the integrations. Moreover a general cover letter providing some clarifications about the reviewers’ remarks has also been submitted.
Sincerely,
Fabrizio D’Amico, Luca Bianchini Ciampoli, Alessandro Di Benedetto, Luca Bertolini and Antonio Napolitano
POINT-BY-POINT RESPONSES
REVIEWER 1:
Point 1: Please suggest the information input to the current pave management system and the requirements for improvement to the BIM-oriented digital model.
Response 1: We want to thank the reviewer for this suggestion and we deeply regret to have not well addressed the main idea of the selected topic for the paper. The case study used for the application described in the paper is not involved in any pave management system, but it was used as an example for the creation of a BIM-oriented “preliminary” digital model, that can be potentially used in a future BIM-based PMS approach. In any case, in order to better clarify the sense of the paper, we partially changed the title, the abstract, the introduction and the conclusions.
Point 2: It is known that it is very difficult to distinguish soil profiles with ground-penetrating radar. Please suggest whether the base layer, sub-base layer, subgrade, and ground under the pavement can be distinguished by GPR.
Response 2: The authors agree with reviewer’s comment. Pavement layering by GPR is no trivial matter, which would require a deeper focus. We had considered to skip this passage as the interpretation of GPR results is here used as an input file for the BIM approach. However, in accordance to the reviewer’s comment, we feel that more information on the reconstruction of pavement structure by GPR would actually raise the comprehensiveness of the paper. Accordingly, some relevant information have been added to section 3.2, along with the integration of Fig. 8 (actually organized in Fig. 8a and b) with an example of layer recognition from a GPR scan.
In particular, in this study, a correct interpretation of the depth of layers interface has been achieved by calibrating the speed of propagation of EM waves within the pavement through various corings extracted along the surveyed stretch.
Point 3: I know that road friction coefficient and soil support value are essential information in the pave management system, but it is necessary to obtain this information with a laser scanner and GRP and explain whether it is possible to input it into the BIM model.
Response 3: Thank you again for your comment and for suggesting the need of a couple of very important and essential values (road friction coefficient and soil support) for characterizing the pavement conditions. In this study, our main effort is to demonstrate the possibility to create a “preliminary” pavement BIM model integrating the results from NDT surveys that it is not strictly correlated with the road friction and soil support. Once a “preliminary” BIM model will be set and ready, multiple information from different measurement campaigns can be inserted such as road friction coefficient and soil support, in order to update the model at different times and according to different instruments used. So, the study doesn’t yet involve this next stage and we will take into account your suggestion for future research activities.
Reviewer 2 Report
The headline states BIM and PMS but the article only slightly touches these two topics. Scanning roads using NDT, putting the data together into a 3D model and geo-reference it does not make a BIM. It is a common miss conception that 3D models in general can be considered as BIM. BIM consist of well defined objects and I hoped to find them within your cases. For instance breaking the scanned road up into 1 or 5 meter long pieces and define them as objects containing attribute data including information coming from the GPR. But this is not what you are presenting. You write about creating parametric sections (line 98-100, 266), but what parametric geometry are you referring to - the length (?) – you are scanning the complete surface? You do not need any parametric geometry to create BIM objects. A B-rep geometric representation of the surfaces and layers beneath is quite sufficient thinking of your MLS and GPR data. Put your road together as a puzzle consisting of scanned road objects.
I do not quite understand why you are using Infraworks in this project, because this is not really a professional point cloud editing tool but rather used for visualizing a collection of data and sketching ideas within infrastructure. Why are you using this odd software?
I really miss some words about the original road design data. Some plans even if only on paper (important statement actually) must exist. There must be some accessible information about the road construction from the time when the road was built. You do not mention anything about accuracy of your scans and survey in general. This should be aligned with the information available in the PMS.
What you show is not new rather mainstream and of the shelf technology. On the other hand the idea of having a PMS which can handle real well defined BIM data and handle ground scanning information together with other data as attribute data is not a new thought but novel if you would be able to show that?
I will not dismiss your article because you want a lot of the right things, you are just not really showing it. So I hope you will continue your work and pursue a really BIM based PMS. :-)
Recommendations:
1. Change the headline to something more appropriate and less populistic
2. I would like the article to dismiss most of the talk about BIM and PMS because that is not really what you are doing. Tell a story about the work and methodology you are demonstrating in the cases. That you gather a lot of data in the field and put them wisely together to describe a road surface and its condition, and that these data are a necessity for a futureBIM based PMS implementation.
3. So if there really is an existing PMS and all the good data sampled in the cases splitting the road up into BIM objects, and thereby providing a BIM model with updated underground data – that could be cool. Having a screen dump from the existing PMS presenting these new data would be even better – what about a mockup – showing your idea of where you want to go. (just an idea). I know that Norway is having something very soon.
4. By a way of introduction I think you need to argue why you need your PMS to be BIM based. Most professionals within the area do not think so. You are also mentioning Digital Twins (DT) (line 23) but BIM-based PMS’ are certainly fra away from a DT. Just leave it.
Line 233: What axis?
Line 239: “adequate level of approximation” what is adequate here – be more precise
Author Response
Dear Reviewer,
we are pleased to submit the new revised version of the paper “Integrating non-destructive surveys into a BIM-oriented digital model for road pavements management” that addressed your useful remarks, and the point-by-point response to your comments.
In fact, based on your comments, we have had the opportunity to examine more in-depth our paper and our work in general. We have considered in depth all the useful comments made by you and by the other reviewers. Below you can find your point-by-point response and, in the new revised paper we have included all the integrations. Moreover a general cover letter providing some clarifications about the reviewers’ remarks has also been submitted.
Sincerely,
Fabrizio D’Amico, Luca Bianchini Ciampoli, Alessandro Di Benedetto, Luca Bertolini and Antonio Napolitano
POINT-BY-POINT RESPONSES
REVIEWER 2:
Point 0: The headline states BIM and PMS but the article only slightly touches these two topics. Scanning roads using NDT, putting the data together into a 3D model and geo-reference it does not make a BIM. It is a common miss conception that 3D models in general can be considered as BIM. BIM consist of well defined objects and I hoped to find them within your cases. For instance breaking the scanned road up into 1 or 5 meter long pieces and define them as objects containing attribute data including information coming from the GPR. But this is not what you are presenting.
Thank you for remarking on this point. We regret to have not well and fully addressed it in the first version of our paper. We appreciated the reviewer’s suggestion to specify the differences between a “simple” 3D model and a BIM model. The model presented in this article is defined only by the information collected during the surveys. As it’s a model of a road it can contain information regarding its condition, such as the materials that form the superstructure or, in a more general way, the direction of travel on the carriageway and other data that may concern an infrastructure. The study presents a process to integrate NDT data into said model, such as the GPR data that can define how the superstructure’s dimensions vary along the infrastructure. So, the result is not only a 3D model composed of the different layers of the superstructure of the road, as the software used can identify it as a road with its own attribute data. Moreover, the model is not divided in smaller pieces containing information, but it is a single object that can still contain also data coming from the NDT surveys.
In the revised paper, in section 3.4, we have added some sentences to specify the characteristics of the model.
You write about creating parametric sections (line 98-100, 266), but what parametric geometry are you referring to - the length (?) – you are scanning the complete surface? You do not need any parametric geometry to create BIM objects. A B-rep geometric representation of the surfaces and layers beneath is quite sufficient thinking of your MLS and GPR data. Put your road together as a puzzle consisting of scanned road objects.
Thank you for highlighting this important lack of information. In the revised paper we have modified the sentence (line 315-318) and we inserted a new one (line 351-353) to specify that the parametric cross-section can adapt its width to the one of the analyzed road and the thickness of its layers to the data provided by the GPR. This comes in the form of three-dimensional polylines that work as targets for the section as it is extruded along the axis of the road.
I do not quite understand why you are using Infraworks in this project, because this is not really a professional point cloud editing tool but rather used for visualizing a collection of data and sketching ideas within infrastructure. Why are you using this odd software?
We are sorry if the explanation about the use of Infraworks is not so clear in the paper. Even though Infraworks is not really a point cloud editing tool, it still allows for the extraction of the needed objects for the model from a certain point cloud, as it can detect different intensity areas within the point cloud. We used Infraworks for the creation of three three-dimensional polylines corresponding to the road markings, as these are subsequently used to determine the position and development of the model, which is created by extruding the parametric road cross-section along the central one of the three. Infraworks can automatically draw these objects along areas of the point cloud that are detected to have different intensity of the returned laser signal than the others around them, allowing for a more efficient and quick way to generate the needed objects for the model. In the revised paper we have partially modified section 3.1 to explain the reason for the use of the Infraworks software.
I really miss some words about the original road design data. Some plans even if only on paper (important statement actually) must exist. There must be some accessible information about the road construction from the time when the road was built. You do not mention anything about accuracy of your scans and survey in general. This should be aligned with the information available in the PMS.
Thank you again for your useful remarks. Unfortunately, we hadn’t any road design data available especially because the study is not involved in a real PMS, actually. Obviously, the design plans can generally help in the construction of a BIM-model of a road. Again, with respect to the “accuracy of your scans and survey in general” some information have been addressed in chapter #3, but these don’t arise from any existing PMS as the study is not based on it.
What you show is not new rather mainstream and of the shelf technology. On the other hand the idea of having a PMS which can handle real well defined BIM data and handle ground scanning information together with other data as attribute data is not a new thought but novel if you would be able to show that?
I will not dismiss your article because you want a lot of the right things, you are just not really showing it. So I hope you will continue your work and pursue a really BIM based PMS. :-)
We want to thank again the reviewer for appreciating the intention of the authors on this important topic of research. We hope that thanks to the revisions provided in the new version of the paper, the main purpose of the study is more clear. We don’t want to apply any PMS or to demonstrate a novel PMS approach, instead we want to create a “preliminary” pavement BIM model integrating the results from NDT surveys as specified in some parts of the paper.
Point 1: Change the headline to something more appropriate and less populistic
Response 1: Thank you for highlighting the need to change the headline. In the revised version of the paper, the title has been changed in “Integrating non-destructive surveys into a preliminary BIM-oriented digital model for possible future application in road pavements management”.
Point 2: I would like the article to dismiss most of the talk about BIM and PMS because that is not really what you are doing. Tell a story about the work and methodology you are demonstrating in the cases. That you gather a lot of data in the field and put them wisely together to describe a road surface and its condition, and that these data are a necessity for a future BIM based PMS implementation.
Response 2: We regret to have not well addressed the main concept of the study in the original version of the paper. We completely agree with your remark. In fact, the main aim of the study is to demonstrate the possibility to create a “preliminary” BIM-model of the infrastructure (with a special focus on the pavement element) starting from NDT surveys. Once the BIM model is set, it is effectively possible to use it for a future BIM-based PMS implementation.
Thanks to your suggestion in the revised version of the paper we have better specified this observations.
Point 3: So if there really is an existing PMS and all the good data sampled in the cases splitting the road up into BIM objects, and thereby providing a BIM model with updated underground data – that could be cool. Having a screen dump from the existing PMS presenting these new data would be even better – what about a mockup – showing your idea of where you want to go. (just an idea). I know that Norway is having something very soon.
Response 3: Thank you again for your remark. As written before, unfortunately the study is not involved in a real PMS actually, but we can work for this!! We specified it in the revised version of the paper.
Point 4: By a way of introduction I think you need to argue why you need your PMS to be BIM based. Most professionals within the area do not think so. You are also mentioning Digital Twins (DT) (line 23) but BIM-based PMS’ are certainly fra away from a DT. Just leave it.
Response 4: Thank you for highlighting the specify a possible to change about our PMS. As mentioned before, we are trying to verify the possibility to create a “preliminary” BIM-oriented digital model by the information from different non-destructive surveys in order to potentially use it in a future BIM-based PMS activities. We think that, at least in Italy, the proposed approach could be well evaluated by the highways concessionaires or, in general, by the management entities that could benefit from a digital model capable of storing data regarding pavement conditions.
Point 5: Line 233: What axis?
Response 5: In the line 233 (line 315 in the revised version) we meant the axis of the cross-section and it has been revised.
Point 6: Line 239: “adequate level of approximation” what is adequate here – be more precise
Response 6: We added lines from 322 to 326 to better specify how the section can approximate the existing ground as described by the point cloud generated by the MLS survey. As the study represents a first process to define a BIM model generated from survey data, we wanted to guarantee the possibility to choose the level of detail with which the surface irregularity was reproduced. Once the needed data is provided, the digital model can then be created with different level of accuracy, as it could be used for different reasons and not only for the analysis of its surface. Thank you for your remark.
Reviewer 3 Report
Dear Authors,
Thank you for submitting your paper to the Infrastructure journal. The paper is presenting some very interesting results. There are some points that need to be improved. I recommend the following comments for improving the quality of presentation and in particular the BIM model.
1- In section 3.3, need more effort to see the accuracy of the parametric elements. I suggest describing the efficiency of flexibility in this section, and how the integration is done by approximation? Is that could be a limitation of accuracy?
2- Figure 12 the digital model, is not clear, I would propose to zoom, and explain in different colours ... different damages, features? I can see the model seems just to see the integration? are there any results to see real-time monitoring? if not please indicate the limitations of the study.
3- The study is very interesting but still it is not mentioned if they used BIM as-built model, or IFC or any other model? please add this and describe reasons for model selection with NDTs data.
4. In conclusion, I cannot see any validation results? will be interesting to describe or at least add something relevant to validation approaches.
5. The paper is mostly related to the integration that findings, it would be interesting at least to show some results.
Thank you!
Author Response
Dear Reviewer,
we are pleased to submit the new revised version of the paper “Integrating non-destructive surveys into a BIM-oriented digital model for road pavements management” that addressed your useful remarks, and the point-by-point response to your comments.
In fact, based on your comments, we have had the opportunity to examine more in-depth our paper and our work in general. We have considered in depth all the useful comments made by you and by the other reviewers. Below you can find your point-by-point response and, in the new revised paper we have included all the integrations. Moreover a general cover letter providing some clarifications about the reviewers’ remarks has also been submitted.
Sincerely,
Fabrizio D’Amico, Luca Bianchini Ciampoli, Alessandro Di Benedetto, Luca Bertolini and Antonio Napolitano
POINT-BY-POINT RESPONSES
REVIEWER 3:
Point 1: In section 3.3, need more effort to see the accuracy of the parametric elements. I suggest describing the efficiency of flexibility in this section, and how the integration is done by approximation? Is that could be a limitation of accuracy?
Response 1: Thank you for your remark, we tried to be more precise in the description of the parametric elements of the model, as also other reviewers suggested we better specified how those elements work. In particular, as reported in Point 6 of reviewer 2 “We added lines from 322 to 326 to better specify how the section can approximate the existing ground as described by the point cloud generated by the MLS survey. As the study represents a first process to define a BIM model generated from survey data, we wanted to guarantee the possibility to choose the level of detail with which the surface irregularity was reproduced. Once the needed data is provided, the digital model can then be created with different level of accuracy, as it could be used for different reasons and not only for the analysis of its surface.”.
Point 2: Figure 12 the digital model, is not clear, I would propose to zoom, and explain in different colours ... different damages, features? I can see the model seems just to see the integration? are there any results to see real-time monitoring? if not please indicate the limitations of the study.
Response 2: Thank you for the remark, we tried to better visualize the model within the figures, even though we are not sure if you are referring to Figure 11 (in which the model is displayed) or Figure 12 (which shows the results of the different analysis that were carried out using said model). The model created allows to integrate different data in a single object, but can also be used to perform some analysis regarding the conditions of the pavement. For instance, in the article results of analysis on the slope of the superstructure’s layers are shown (Fig.12a and b), which can highlight areas of the road that may need to be more thoroughly analyzed, as they can represent a criticality for the infrastructure. In particular, figure 11 and 12 show the same model but with different visualization styles, as the former shows the model itself with the different materials that make up the superstructure, while the latter shows the model colored by the results of the analysis performed on it. We are aware of the fact that these represent only a first approach at the management phase of road pavements, but we hope to continue our work on the matter to improve the process hereby presented.
Point 3: The study is very interesting but still it is not mentioned if they used BIM as-built model, or IFC or any other model? please add this and describe reasons for model selection with NDTs data.
Response 3: We want to thank again the reviewer for this suggestion. As mentioned in different previous responses, we didn’t use any BIM as-built model, and neither an IFC, as we a trying to create a new BIM-model. This model is created by relying only on the results of survey carried out on site, as no other information was used to recreate the analysed road. Thanks to your useful remark, this information has been addressed in line 364-368.
Point 4: In conclusion, I cannot see any validation results? will be interesting to describe or at least add something relevant to validation approaches.
Response 4: For this study the main objective was to integrate different data regarding road pavement in a single new digital model. The validation phase of the modeling that was carried out was not possible so far, as it would require the comparison of the real infrastructure with its digital representation, and it was decided to postpone it for future integrations of the work hereby presented. However, as a partial validation of the geometric reconstruction of the pavement, the depth of layers interface has been achieved by calibrating the speed of propagation of EM waves within the pavement through various corings extracted along the surveyed stretch, as it is now stated in Section 3.2. This operation ensured the limitation of potential misinterpreation of pavement layering by GPR.
Point 5: The paper is mostly related to the integration that findings, it would be interesting at least to show some results.?
Response 5: As we a trying to create a new BIM-model, the first result is represented by the model itself as it can integrate the different data provided by the surveys. At this stage, the integration between the result provided by NDTs surveys seems to be possible and effective for the main purpose of the study. The next stage of the research provides further integrations and possible model updates.